# A Revised Temperature-Dependent Remineralization Scheme for the Community Earth System Model (v1.2.2)

Elizabeth K. Brabson[1], Loren F. Doyle[2], R. Paul Acosta[2], Alexey V. Fedorov[1], Pincelli M. Hull[1], Natalie J. Burls[2]

[1]Earth & Planetary Sciences, Yale University, New Haven, CT, 06520 USA
[2]Oceanic, Atmospheric, & Earth Sciences, George Mason University, Fairfax, VA, 22030 USA

*Correspondence to*: Elizabeth K. Brabson (liz.brabson@yale.edu)

**Abstract.** Export of carbon from the euphotic zone to intermediate and deep water plays a critical role in the ocean's feedback response to a warming climate. However, as water temperature increases so does the rate of bacterial respiration at the base of the biological pump, resulting in more efficient recycling of carbon in the upper ocean, less efficient export of carbon to depth, and a diminished net negative feedback on climate. Therefore, to better predict climate response associated with changes in ocean carbon storage in warming scenarios, it is imperative to incorporate temperature-sensitive mechanisms, such as bacterial respiration (remineralization), into Earth system models. Here, we employ a new temperature-dependent parameterization for remineralization (Tdep) in the Community Earth System Model version 1 (CESM1) applied to gravitationally sinking particulate organic carbon (POC) in a preindustrial control simulation. We find that the inclusion of Tdep in both low and high-resolution model configurations more accurately captures regional heterogeneity in POC transfer efficiency while preserving the overall trends in nutrient distribution and attenuation of sinking particulate matter when compared with modern empirical data. Inclusion of this parametrization will allow for improved predictions of temperature-sensitive mechanisms impacting carbon storage in the warming ocean.

## 1 Introduction

The biological carbon pump can serve as a net negative feedback on the global carbon cycle through the export of organic carbon to the ocean's interior where it can be sequestered from the ocean-atmosphere interface for hundreds to thousands of years. Today, an estimated 10 Pg C yr$^{-1}$ is exported as gravitationally sinking particulate organic carbon (POC) from the upper ocean resulting in the storage of approximately 1300 Pg C in the ocean's interior (Nowicki et al., 2022). The extent to which this will continue in warmer future oceans remains to be fully constrained, partially due to differences in the modeled response of the marine carbon cycle to warming. Recent climate model intercomparisons, for example, using an SSP3-7.0 warming scenario conducted as part of the Coupled Model Intercomparison Project Phase 6 (CMIP6) (Eyring et al., 2016) show broad agreement in the overall trends of anthropogenic carbon uptake by the global oceans (Melnikova et al., 2021), with decreases in storage as atmospheric $CO_2$ increases. Yet the same models differ significantly in the magnitude and duration of predicted carbon storage by 2100, with decreases in globally averaged export flux at 100 meters ranging from 1.5% to 14.4% (Wilson et al., 2022). Varied approaches across modeling groups in the parameterization of temperature-sensitive feedback mechanisms, such as stratification, ballasting, ecological shifts, circulation, and metabolic processes (e.g., Henson et al., 2022; Boyd, 2015; Keeling et al., 2010; Plattner et al., 2001) account for the lack of consensus amongst models. Therefore, the inclusion and refinement of these key processes is of particular importance for improving predictions of the marine carbon cycle response to future warming. Here, we focus on one of these key factors, namely temperature-sensitive metabolic processes within the biological carbon pump.

The rate of bacterial respiration at the base of the biological pump, also referred to as remineralization, increases with rising temperature, thereby affecting export production (e.g., Boscolo-Galazzo et al., 2021; Marsay et al., 2015; Laws et al., 2000).

In warmer water, the rates of respiration and photosynthesis both increase, with the rate of respiration increasing more rapidly than the rate of photosynthesis (Allen et al., 2005; Brown et al., 2004). This creates an imbalance whereby remineralization reactions can outpace production, resulting in an increase in the net recycling rate of organic material in the upper ocean and a shallowing of the depth of remineralization and associated oxygen minimum zone (e.g., Marsay et al., 2015; John et al., 2014). This lever, temperature-dependent remineralization (Tdep), has been shown to contribute to diminished POC export flux during times of warmth (e.g., John et al., 2014), making it a critical parameter for future warming simulations. However, less than half of CMIP6 models, including the Community Earth System Model (CESM), incorporate this key mechanism (Henson et al., 2022). While other modeling groups have successfully incorporated Tdep (e.g., Stock et al., 2020; Laufkötter et al., 2017; Aumont et al., 2015), distinct approaches to productivity, community structure, particle sinking rates, and aggregation for each model configuration (e.g., Burd, 2024), limit the portable nature of code across models, making each implementation model-specific. Here, we focus on the addition of new model code for Tdep within the CESM1.2 framework.

Biological pump efficiency, or transfer efficiency, is a metric commonly used to quantify changes in export production and is calculated as the percentage of POC flux at 100 meters depth that reaches 1000 meters (e.g., Wilson et al., 2022; Crichton et al., 2021; Stock et al., 2020; Laufkötter et al., 2017). An efficient pump, therefore, is capable of exporting larger volumes of POC to the deep ocean, whereas an inefficient pump recycles more of that carbon in the upper ocean. Statistically derived regional patterns of transfer efficiency based on a data-constrained modeling approach show greater pump efficiency in the cooler higher latitudes and high productivity upwelling regions and lower pump efficiency in the central gyres (Weber et al., 2016) (Fig. 1). Temperature-dependent remineralization is one of several factors influencing this distribution (Cram et al., 2018; Weber et al., 2016). However, due to differences in individual parametrizations across models, these observed regional patterns are not consistently recreated across existing earth system models, even when comparing preindustrial control simulations (Wilson et al., 2022). For example, in five CMIP6 models, the distribution and scale of transfer efficiency deviates greatly from the expected geographic distribution based on observations, with some models predicting generally globally uniform transfer efficiencies of ~10% or less with minimal latitudinal variation, while others predict stronger regional differences between upwelling regions and central gyres yet have little to no latitudinal variation (Wilson et al., 2022). These geographic disparities in modeled transfer efficiency are problematic, particularly when forecasting marine carbon cycle response in projected warming scenarios as mentioned above, as they reflect underlying differences across models and highlight mechanistic uncertainties in our understanding of the marine carbon cycle. Compounded with the fact that surface ocean warming is expected to be regionally heterogeneous, with higher latitudes seeing greater warming relative to lower latitudes (e.g., Cheng et al., 2022; IPCC, 2023), the inclusion of temperature-sensitive feedbacks is critical to capturing regionally differing responses to this warming.

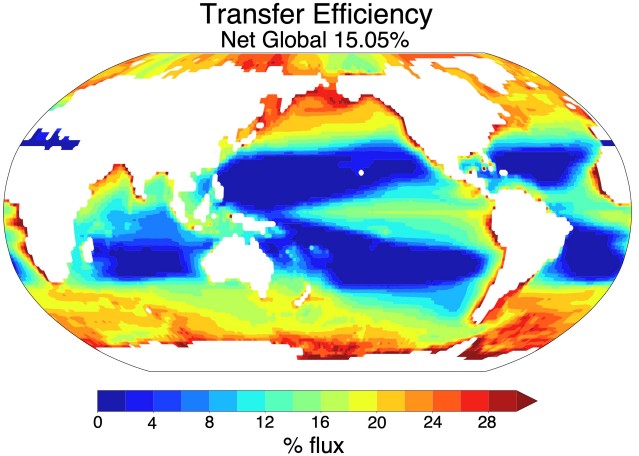

**Figure 1. Data-constrained map of global transfer efficiency.** Figure recreated using data provided by Weber et al., 2016 via personal communication, with permission.

It is clear that the integration of a temperature-dependent remineralization scheme would add an essential feedback on biological pump efficiency that could improve modeled predictions of the global carbon cycle in future warming scenarios. Here, we address this gap through the implementation of a new temperature-dependent remineralization scheme in the Community Earth System Model (CESM v1.2.2) in a preindustrial control simulation. We use a compilation of measured oceanographic data to evaluate the effects of the new parameterization on pump efficiency, both globally and regionally.

## 2 Methods

### 2.1 Model Framework

We modified the ecosystem module of the Community Earth System Model (CESM) version 1.2.2 to include a new temperature-dependent remineralization scheme and examined the effects of these modifications relative to preindustrial control simulations in both low and high-resolution versions of the model. CESM version 1.2.2 includes physical model components representing the atmosphere, land, sea ice, land ice, and ocean and is designed for optimizing computational resources for paleoclimate studies (Hurrell et al., 2013; Shields et al., 2012). The atmospheric component utilizes the Community Atmospheric Model 4 (CAM4) with a low-resolution spectral truncation of T31 (~300 km) and a high-resolution finite volume 2° (~200 km), while the ocean component uses the Parallel Ocean Program version 2 (POP2) with a vertical resolution including 60 layers ranging from 10 meters in the upper 150 meters to 250 meters at abyssal depths. The low-resolution simulations have a horizontal resolution varying from about 3° near the poles to 1° at the equator, while the high-resolution simulations have a 1° global resolution. New low-resolution simulations were branched from year 2500 of a previously equilibrated preindustrial run (Burls and Fedorov, 2014), while the new high-resolution simulation used out-of-the box preindustrial run boundary conditions (Jahn et al., 2015). Both runs were extended to 300 years, which we found sufficient for the ocean model to reach quasi-equilibrium. Results discussed below are reported as averages of the final 30 years of each simulation.

The biogeochemical component of CESM version 1.2.2 tracks physical and biological processes influencing the marine carbon cycle (Moore et al., 2013). The carbon isotope module, which traces $^{13}$C, was recently added (Jahn et al., 2015) and includes standard carbon cycle processes with isotopic fractionation during gas exchange and photosynthesis based on empirical relationships (Zhang et al., (1995) and Laws et al., (1995), respectively). Version 2.12 of the ecosystem module includes zooplankton, as well as three explicit phytoplankton functional groups (diatoms, diazotrophs, smaller phytoplankton) and one implicit group (coccolithophores) (Moore et al., 2013, 2004, 2002; Doney et al., 1996). Primary production can be nutrient, light, and temperature limited and varies as a function of location and functional group. A ballasted fraction of soft particulate carbon is also accounted for using a quantitatively associated ballast scheme for carbonate, silicate, and dust (Armstrong et al., 2002). A byproduct of ecosystem processes, detrital carbon is placed into two general categories for the purposes of remineralization within the ecosystem module, non-sinking and sinking pools (Moore et al., 2002). The non-sinking pool is comprised of nano to pico-sized detrital carbon, also referred to as dissolved organic matter (DOM), and is remineralized more rapidly using a different scheme than the sinking pool. The remaining detrital organic material is placed into the large detrital carbon pool, also referred to as particulate organic carbon (POC), which is subject to gravitational sinking and varying rates of remineralization which are established locally using the scheme outlined in Section 2.2.

### 2.2 Remineralization

Detrital carbon can enter each pool via several pathways (Fig. 2). Zooplankton mortality and phytoplankton aggregation are routed directly to the large detrital pool, whereas phytoplankton non-grazing and grazing mortality are routed to a generalized

detrital carbon pool. Organic material within the detrital carbon pool is partitioned between the large and small detrital pools using a series of parameterizations based on phytoplankton functional group. Particulate carbon that reaches the large detrital pool is further separated to include both a ballasted fraction and an 'excess' fraction, with a prescribed portion of all new productivity being assigned to quantitatively associated (QA) ballast minerals (silicate, carbonate, dust). This portion is essentially protected from the standard remineralization of the non-QA pool, resulting in a diminished amount of remineralization of this material in the euphotic zone and longer remineralization scalelength overall. The remainder of the large detrital pool is considered 'excess' particulate organic carbon ($POC_E$). Remineralization of this pool is a function of local environmental conditions, and this is where the new temperature-dependent remineralization scheme has been applied. Although processes related to the remineralization of DOM and the QA fraction play an important role in the cycling of carbon, they are not the direct focus of this study. Instead, we have focused specifically on the gravitationally sinking flux of POC that is free and available for remineralization, which comprises the majority of the total organic carbon exported from the euphotic zone (e.g., Siegel et al., 2023 and references therein).

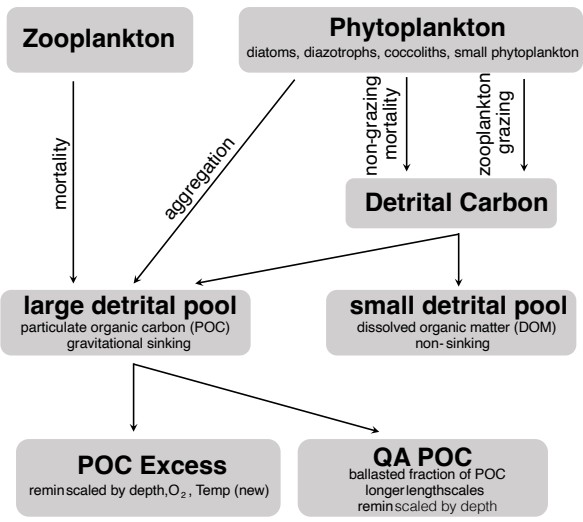

**Figure 2. Source of excess POC in the Ecosystem Module.** The new Tdep scheme is applied to the POC excess pool. The small detrital pool undergoes rapid remineralization and is not addressed in this study.

Attenuation of POC with depth has traditionally been modeled as a power law function, which was originally developed based on particle trap data in the northeast Pacific (Martin et al., 1987). However, this formulation has been shown to vary regionally, which can be accounted for using differing values of the exponential 'b' (e.g., Marsay et al., 2015; Henson et al., 2012). The dependence upon a reference depth in the so-named 'Martin Curve' has also raised concern, particularly when applied across the global domain (e.g., Buesseler et al., 2020). Another approach to quantifying the attenuation of POC flux utilizes an e-folding decay scheme which employs local depth and a remineralization scalelength (e.g., Marsay et al., 2015; Sarmiento and Gruber, 2006; Armstrong et al., 2002), while other schemes directly calculate losses due to remineralization of POC as a function of a prescribed remineralization rate (e.g., Yool et al., 2013; Crichton et al., 2021). The ecosystem module (Moore et al., 2013; Armstrong et al., 2002; Moore et al., 2002; Doney et al., 1996) in CESM1.2.2 calculates the remineralized fraction of sinking POC (non-ballasted portion) by first calculating the flux of excess POC out of the model cell ($POC_{flux\_out}$) using an e-folding decay term ($decay_{POC\_E}$) that is scaled by a dissociation scalelength (poc_diss) (Eq. 1). Mass balance is then applied to implicitly determine the remineralized fraction as shown in Eq. (2). Flux into each cell is equivalent to the flux out of the cell above, and new primary productivity within each cell available for remineralization ($POC_{prod\_avail}$) is also taken into account.

$$POC_{sflux_{out}} = (POC_{sflux_{in}} * decay_{POC_E}) + (POC_{prod_{avail}} * [(1 - decay_{POC_E}) * poc\_diss]) \qquad (1)$$

$$\text{remin} = POC_{flux_{in}} - POC_{flux_{out}} \tag{2}$$

The adjusted Tdep parameterization was included by revising the e-folding decay term (*decay_POC_E*) (Eq. 3), as described below. The original remineralization scheme within CESM relied upon a well-tuned scalelength parameter, *scalelength_parm*, within the decay term that increased from a base dissociation scalelength of 88 meters according to local depth. While this function adequately captured the net attenuation of POC flux in the original preindustrial control runs, it did not capture the expected regional distribution of transfer efficiency, with higher latitudes exhibiting efficiencies between 10 to 18%, approximately 10% lower than anticipated values in these regions (see, e.g. Fig. 4 and discussion below). With the addition of the new Tdep parameterization, the original scalelength parameter tuning was modified to include an increase in the base dissociation scalelength to 150 meters, slight adjustment of the scalelength depth bands of *scalelength_parm*, and a decrease by roughly half for the weighting of the scalelength with each depth band (Table 1). To account for the role of temperature on the remineralization scalelength, a new temperature parameter for POC, *k_temp_poc* (Eq. 4), was incorporated which includes a constant parameter, *k*, with a prescribed value of 0.05, which falls into the middle of the highest confidence range for the global domain as examined in the Laufkötter et al., 2017 study, who have a similar yet slightly different formulation for temperature-dependence. The new temperature parameter takes into account the local water temperature, *TEMP*, allowing for temperature controls on remineralization to be unique to the temperature within each grid cell. At a reference temperature of 20°C, this *k* value of 0.05 is equivalent to a $Q_{10}$ of 2.72. All new parameterizations were fit sequentially and optimized using a compilation of measured POC flux data and global transfer efficiencies aligned to Weber et al., 2016.

$$\text{decay}_{POC_E} = \exp\left[\frac{(-dz_{loc}) * (k_{temp_{POC}})}{150 * (scalelength_{parm}) * (O_{2_{parm}})}\right] \tag{3}$$

Where:

$$k_{temp_{POC}} = \exp[k * TEMP] \tag{4}$$

| Original | | Revised Tdep | |
|:---:|:---:|:---:|:---:|
| depth (m) | scalelength parameter | depth (m) | scalelength parameter |
| 130 | 1.0 | 150 | 1.0 |
| 290 | 3.0 | 400 | 2.0 |
| 670 | 5.0 | 1000 | 3.5 |
| 1700 | 9.0 | 2000 | 4.5 |

**Table 1. Scalelength code modifications.** Original and revised depth bands and scalelength values.

## 2.3 Model Evaluation

The effects of temperature-dependent remineralization on export flux of POC are primarily evaluated using transfer efficiency and POC flux attenuation. Transfer efficiency is calculated by dividing the modeled POC flux at 985 meters depth by the flux at 100 meters and multiplying the result by 100 to obtain a percentage. Global patterns of modeled transfer efficiency are then

compared to data-constrained efficiencies reported by Weber et al., (2016) who define transfer efficiency as the amount of POC leaving the base of the euphotic zone, measured at 100 meters, that reaches at depth of 1000 meters. It is important to note that the depth of the euphotic zone is not globally homogeneous (e.g., Wu et al., 2021), therefore the characterization of its use for transfer efficiency is somewhat imprecise, which is why 100 meters depth was chosen for calculated transfer efficiency in this study.

In order to assess the attenuation of POC flux with depth, a compilation of measured POC flux values were used from specific sites around the globe reflecting a variety of temperature, circulation, and productivity conditions (Fig. 3 and Table 2). As the compilation consists of data obtained from differing depths and techniques and are reported using different unit conventions, all POC flux data were normalized to either measured or estimated flux at 150 meters. For sites where flux was not explicitly measured at 150 meters, an exponential function was fit to the existing data and used to extrapolate estimated flux at 150 meters.

Nutrient distribution in the upper ocean is also assessed, as enhanced remineralization increases net nutrient content in the upper ocean. Upper ocean phosphate content for the preindustrial control and Tdep simulations was compared with phosphate data reported in the World Ocean Atlas 2023 (WOA23). All CESM simulations were regridded to a 1x1 degree horizontal resolution for direct comparison with WOA23 data which were retrieved at the same 1x1 degree grid (Reagan et al., 2023). For consistency, regional assessments of upper ocean nutrient content were completed using the 1x1 degree WOA05 basin mask for both modeled and measured values. Reported global values are calculated as a 'five basin' summative total for the Atlantic, Pacific, Indian, Southern and Arctic Oceans. Correlation ($R^2$) and centered Root Mean Square Error (cRMSE) calculations were completed using Python resources at NCAR.

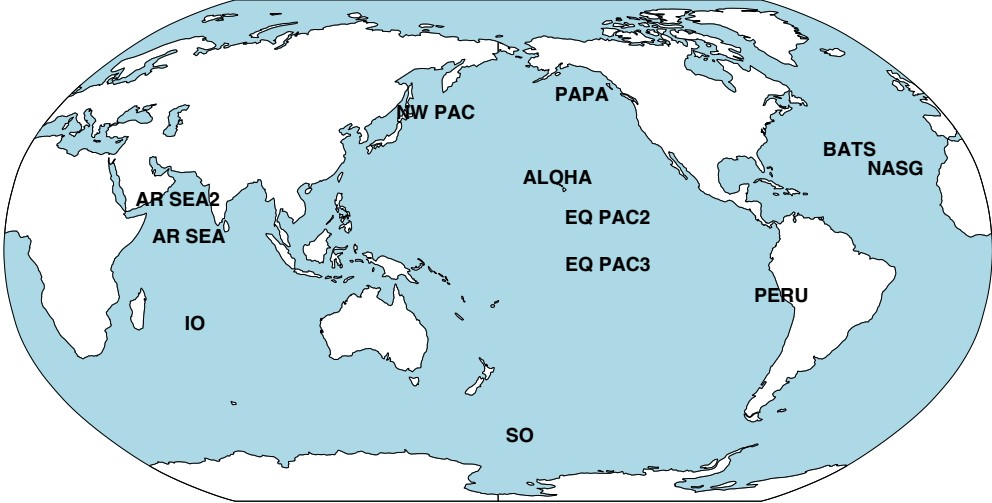

**Figure 3. Map illustrating specific site locations used for the POC compilation.** See Table 2 below for site-specific information.

| Site | Latitude | Longitude | Reference |
|---|---|---|---|
| AR SEA | 4 N | 067 | Seo, 2024 |
| AR SEA2 | ~16 N | 061.5-062.0 | Henson et al., 2012; Lutz et al., 2007 |
| IO | 24 S | 067 | Seo, 2024 |
| NW PAC | 44 N | 155 | Henson et al., 2012; Lutz et al., 2007; Kawakami and Honda, 2007 |
| PAPA | 50 N | 215 | Wong et al., 1999 |
| ALOHA | ~23 N | 202 | Buesseler et al., 2008 |
| EQ PAC2 | 9-11 N | 220 | Henson et al., 2012; Lutz et al., 2007 |
| EQ PAC3 | 5 S | 220 | Henson et al., 2012; Lutz et al., 2007 |
| PERU | 15.5 S | 284 | Martin et al., 1987 |
| SO | ~ 60-63 S | ~ 190 | Henson et al., 2012; Buesseler et al., 2003; Collier and Dymond, 1994; Honjo and Dymond, 1994 |
| BATS | ~ 32 N | ~ 314 | Lutz et al., 2002 |
| NASG | ~ 26 N | ~ 329 | Marsay et al., 2015 |

**Table 2. POC flux compilation.** Site abbreviations, latitude, longitude, and data source.

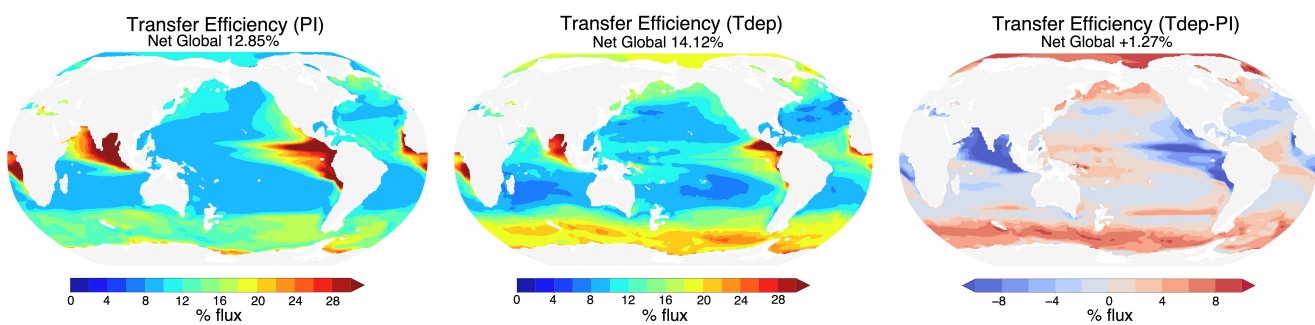

**Figure 4. Global transfer efficiency.** Modeled transfer efficiency for preindustrial (PI) control (left) and Tdep simulations (center). Differences with the inclusion of Tdep are shown in the right panel.

## 2 Results

Results for both low and high resolution configurations with Tdep are in broad agreement, as such, only high resolution results are discussed here, with several small distinctions for low resolution results being highlighted in the Supplemental section as noted below. The inclusion of temperature-dependent remineralization results in new regional patterns of transfer efficiency that are in better alignment with those predicted by Weber et al., (2016) when compared with the preindustrial control simulation without temperature-dependent remineralization (Fig. 4). Cold high-latitudes see the most significant change, with efficiencies increasing from 2 to 8% in the Southern Ocean and North Pacific. Regions of strong upwelling and associated high productivity in the eastern equatorial Pacific and Atlantic experience decreases in efficiency of around 2-8%, yet net global transfer efficiency increases only 1.27% with the inclusion of Tdep. Zonally averaged transfer efficiencies across all longitudinal bands show a net increase in transfer efficiencies north and south of approximately 40° and a net decrease in efficiency equatorward of these latitudes with modest 1-$\sigma$ interannual variability over the final 30 years of the simulation analysed here (Fig. 5). Of the net modeled change, lower latitudes between 30°S and 30°N account for approximately a 22% decrease, while latitudes higher than 30° in both the north and south represent increases of ~51% and 26%, respectively.

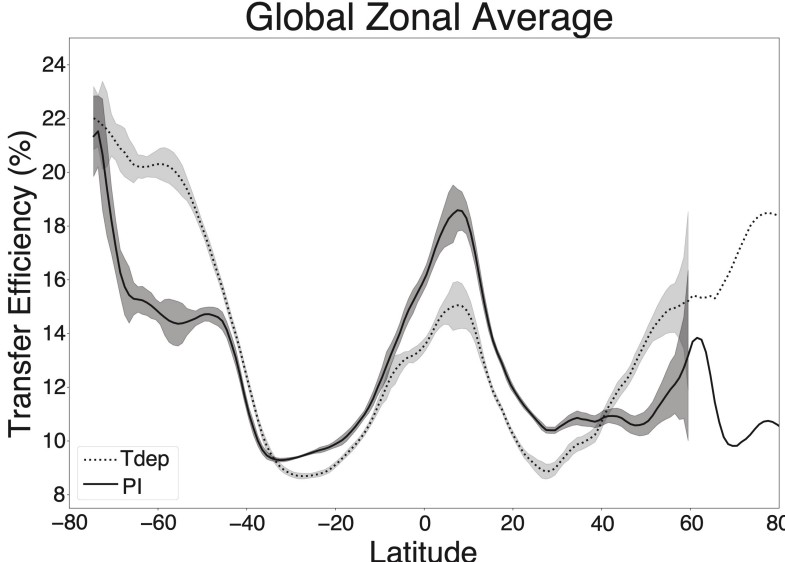

**Figure 5. Global zonal average transfer efficiency.** Preindustrial control results are shown as a solid line, and Tdep results as a dashed line. Interannual variability for the final 30 years of the run analysed here is shown as shaded 1-σ bands. The highest interannual variability occurs in the highly productive Southern Ocean, Equatorial Upwelling regions, and high northern latitudes. 1-σ bands are not shown north of 60N, where internal variability exceeds the range of transfer efficiency displayed due to sea ice impacts on primary productivity.

Modeled transfer efficiency also responds differently to Tdep at the basin-scale, with several notable differences between the low and high-resolution simulations (see Supplemental for discussion of low-resolution results). The Indian and Arctic Oceans together account for ~73% of the net change (45% and 28%, respectively), followed by the Southern Ocean at 20%, Atlantic Ocean at 8%, and Pacific Ocean at 1% (Fig. 6). The Atlantic Ocean experiences a net positive shift poleward of 45°N and a negative shift between 15°N and 45°N, while the Pacific has similar a similar configuration but larger magnitude, particularly in the Northern Pacific where efficiencies increase by up to 5%. Both the Arctic and Southern Oceans have an increase in efficiency of roughly 1-8%. Across equatorial bands, the Atlantic, Pacific, and Indian Oceans all have decreases in efficiency with the inclusion of Tdep, with the strongest change of around 10% taking place in the Indian Ocean. Overall, all basins experience a net decrease in transfer efficiency in the lower latitudes and net increase in higher latitudes when zonally averaged.

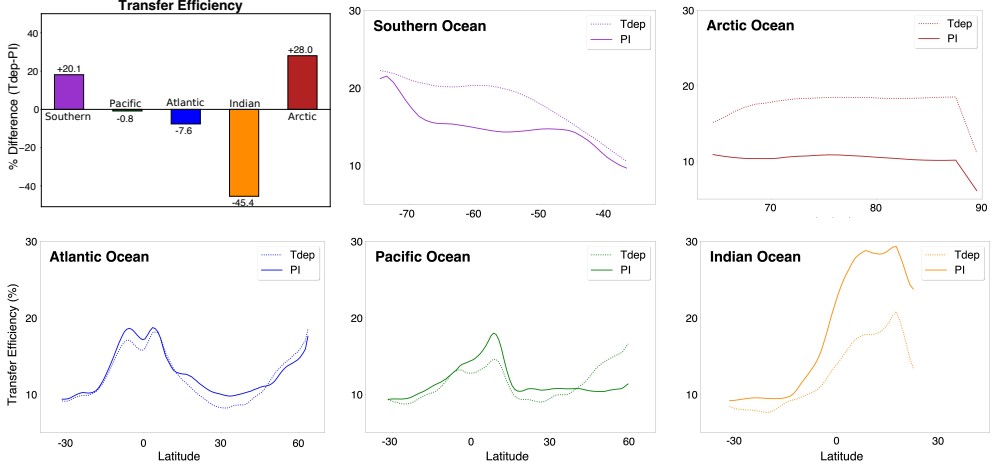

**Figure 6. Basin-scale transfer efficiency.** Zonally averaged transfer efficiency per basin (color) for preindustrial control (solid lines) and Tdep (dashed lines) simulations.

The attenuation of POC flux with depth is in broad agreement with measured values for most reported sites, with subtle differences dependent upon regional conditions (Fig. 7). For warm water locations, such as BATS, NASG, and the Equatorial Pacific, flux attenuation is similar or slightly improved versus preindustrial control, whereas cold water locations of PAPA and NWPac have a roughly equivalent flux attenuation as the preindustrial control run (Fig. 7). The Southern Ocean site performs slightly worse at the 1000 m depth interval with regards to POC flux attenuation, yet the resulting increase in transfer efficiency in this same region is in better alignment with anticipated values. At the same time, the PAPA site in the Northern Pacific has POC flux attenuation that performs better with the introduction of Tdep while also resulting in an improved increase in transfer efficiency. These site-specific differences in the cold water regions could result from regional differences in community structure which impact particle size and aggregation, both of which are not accounted for in the gravitationally sinking POC of this BGC module. The largest divergence from measured POC flux is found in the high productivity regions off the coast of Peru and in the Arabian Sea, which could also be influenced by the secondary parameterizations and will be discussed further below. Most sites have similar or decreased flux below 150 m in the Tdep simulations versus preindustrial control, with the exception of the high productivity sites at PERU, Arabian Sea, EQPAC2, and NWPac (Fig. 7).

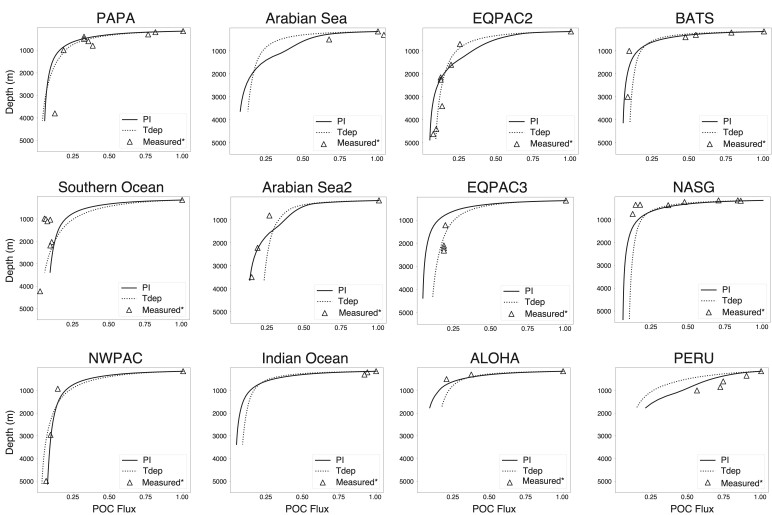

**Figure 7. Site-specific POC flux attenuation.** Measured values are denoted with open triangles (*see Table 2 for source references). As in previous figures, solid lines are preindustrial control and dashed lines are Tdep. All flux attenuations are normalized to POC flux at 150 meters.

Upper ocean phosphate distribution displays equivalent negative biases versus WOA23 data of 0 to 1.0 mmol/m$^3$ in both the high northern and southern latitudes when compared with the preindustrial control simulation (Fig. 8). Most of the tropics and subtropics have enhanced nutrient content versus the control, most notably across the central and western equatorial Pacific and Indian Oceans. Upwelling regions in the eastern equatorial Pacific remain fairly consistent with control biases, while the Arabian Sea region shifts from a slightly negative to a slightly positive bias. The Bay of Bengal also has a slight increase in the positive bias. At the basin scale, both the PI Control and Tdep simulations for the Atlantic perform well compared with WOA data, with $R^2$ values slightly above 0.8, while the Tdep simulation has a slightly higher cRMSE value by about 0.02, but well within reasonable error. In the Southern, Indian, and Arctic Oceans, both simulations perform almost identical, while the largest difference between simulations is seen in the Pacific basin, where the Tdep simulation performance dips slightly versus the PI control by about 0.1 for $R^2$ and 0.05 for cRMSE (Fig. 9) likely driven by the enhanced nutrient content in the tropics as discussed further below. It is important to note that published comparisons between the World Ocean Atlas (WOA) and GLODAP indicate a characteristic inter-product mismatch in global ocean phosphate concentrations of approximately 0.03 μmol/kg when depth-averaged over the full water column (Garcia et al., 2024). This value reflects inter-product disagreement

rather than formal uncertainty in any single dataset. For consistency with the model diagnostics reported here, this mismatch scale is converted to ≈0.03 mmol/m³ assuming a representative seawater density of 1025 kg m⁻³. Accordingly, differences in model–WOA cRMSE on the order of this value are interpreted as not meaningfully distinguishable given observational product disagreement.

Control and Tdep simulated upper ocean phosphate trends are similar, with both slightly higher than WOA23 measured values with the exception of similar negative trends in the Northern Pacific across both model simulations (Fig. 10). When averaged zonally per basin, there is broad agreement between the control and Tdep offset values versus WOA23 in the high latitude regions, however the inclusion of Tdep increases the offset values at most locations within the lower latitudes by around 0.2 to 0.3 mmol/m³ (Fig. 10).

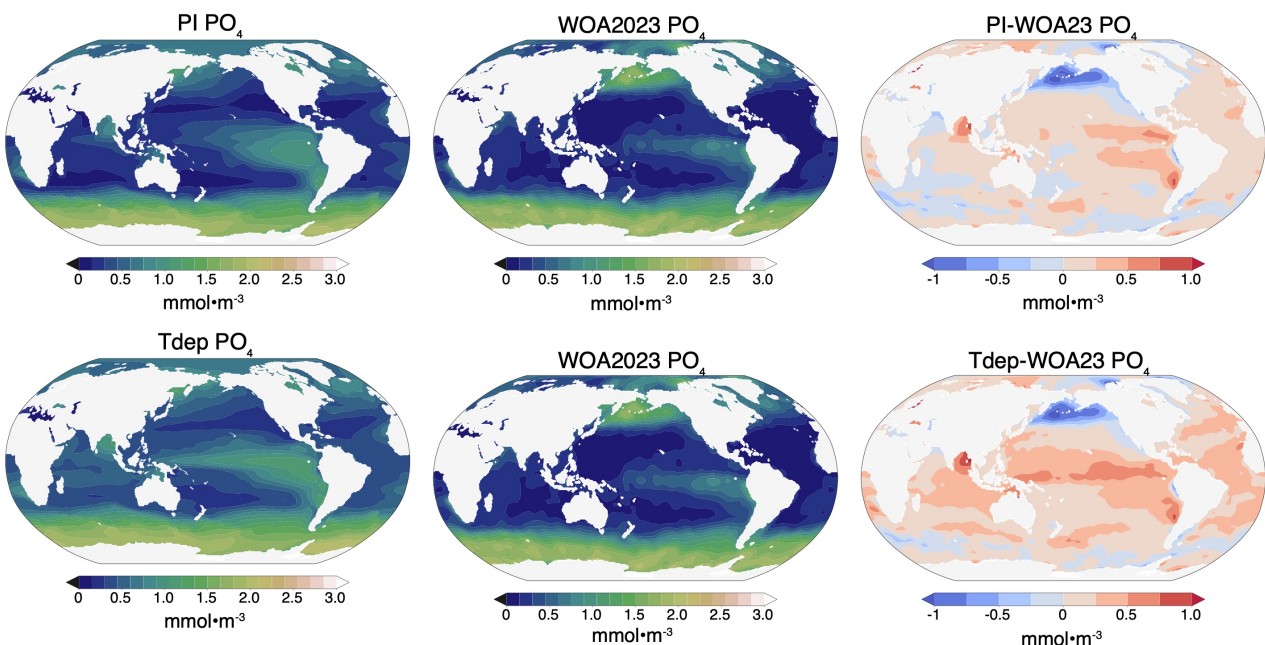

**Figure 8. Upper ocean phosphate.** Reported concentrations of the upper most ocean layer. Left columns are simulated values for preindustrial (PI) and Tdep, center columns are measured World Ocean Atlas data (WOA23), and the right column is model bias.

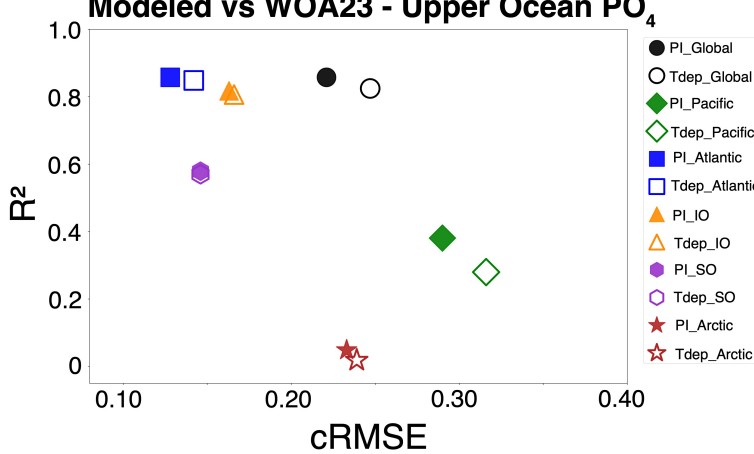

**Figure 9. Basin-scale evaluation of upper ocean phosphate content.** Centered root mean square error (cRMSE) plotted versus correction ($R^2$) for the five largest basins and the 'global' five basin total. Solid symbols represent preindustrial control simulation, while unfilled symbols are Tdep simulations. Color is basin-specific as indicated.

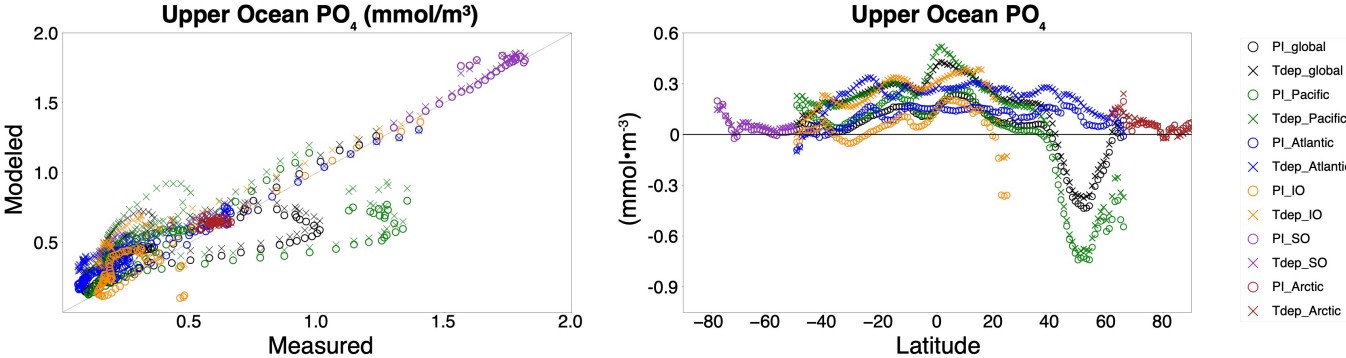

**Figure 10. Data-model comparison.** Upper ocean phosphate content (left panel) as in Figure 8 for preindustrial control (O's) and Tdep (X's) simulations versus World Ocean Atlas (WOA23). Difference between modeled upper ocean nutrient content and WOA23 upper ocean nutrient content (right panel) by latitude. Color is basin-specific as indicated.

## 4 Discussion

The inclusion of temperature-dependent remineralization within the ecosystem module results in improved alignment of basin-scale patterns of transfer efficiency when compared with empirically derived patterns. Region definitions shown in Figure 11 follow Weber et al. (2016), which take into account mechanistic controls on transfer efficiency. Here, all defined regions are retained to ensure a complete representation of global basin-scale responses. Simulations with temperature-dependent remineralization show improved alignment with data-constrained transfer efficiency for all regions except the Subtropical Atlantic (STA) and Subtropical Pacific (STP), which performed approximately equivalent, when compared with preindustrial control simulations (Fig. 11). High productivity regions of the Antarctic Zone (AAZ) and Eastern tropical Pacific (ETP) show the greatest improvement. While net global transfer efficiency increases by a modest 1.27% with the inclusions of the Tdep configuration (Fig. 4), differing meridional and basin-scale adjustments highlight the variable ocean response of remineralization and carbon export to local temperature. Cold high latitudes and high productivity regions see the strongest changes versus preindustrial control simulations, with higher zonally averaged efficiency poleward of 35°S and 40°N and lower efficiency within this latitudinal band. The Southern and Arctic Oceans account for a combined 56% of the increase in transfer efficiency, with a small contribution of this increase coming from the Atlantic and Pacific Basins north of 40-50°N. Higher transfer efficiency in the cold higher latitudes is a direct result of water temperature, with the new parameterization scheme decreasing the rate of remineralization in the cold upper ocean, allowing for stronger export of carbon from the upper ocean to depth.

Decreased transfer efficiency is most notable in high productivity regions of the Eastern Equatorial Pacific and Northern Indian Ocean, with decreases reaching as high as 10%. As anticipated, warmer surface water increases remineralization in the tropical regions, resulting in more shallow remineralization and a shoaling of the oxygen minimum zone (OMZ), shown as an increase in apparent oxygen utilization centered between 50 and 150 meters (Fig. 12a). An associated increase in phosphate in the upper 100 meters (Fig. 12b) drives enhanced primary production and sinking POC flux above 100 meters, centered at roughly 50 meters depth (Figs. 12c & 12d). The net effect of Tdep in the tropics therefore is decreased POC flux at 100 meters and also at 1000 meters, resulting in the net decrease in modeled transfer efficiency, in spite of enhanced productivity in the upper 50 meters. This result brings to question the continued use of fixed depth flux values within the community for calculating transfer efficiency, particularly since the depth range of the euphotic zone and oxygen minimum zones are not globally

uniform. Further discussion of an improved metric is encouraged, such as the use depth-averaged POC flux for the upper 100 meters versus the exact flux at 100 meters depth.

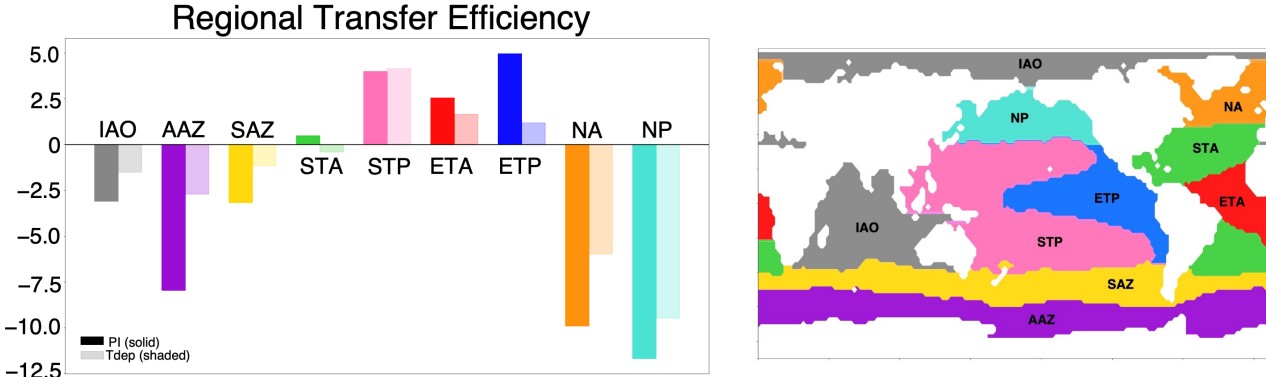

**Figure 11. Regionally averaged transfer efficiency.** Difference of PI Control (solid) and Tdep simulation (shaded) regional transfer efficiency versus data-constrained regional averages defined by Weber et al., 2016 (left). In the right panel, region denoted by color including: NP (North Pacific), STP (Sub-Tropical Pacific), ETP (Eastern Tropical Pacific), SAZ (Sub-Antarctic Zone), AAZ (Antarctic Zone), NA (North Atlantic), STA (Sub-Tropical Atlantic), ETA (Eastern Tropical Atlantic), and IAO (Indian and Arctic Oceans). Figure generated using data provided by Weber et al., 2016 via personal communication, with permission.

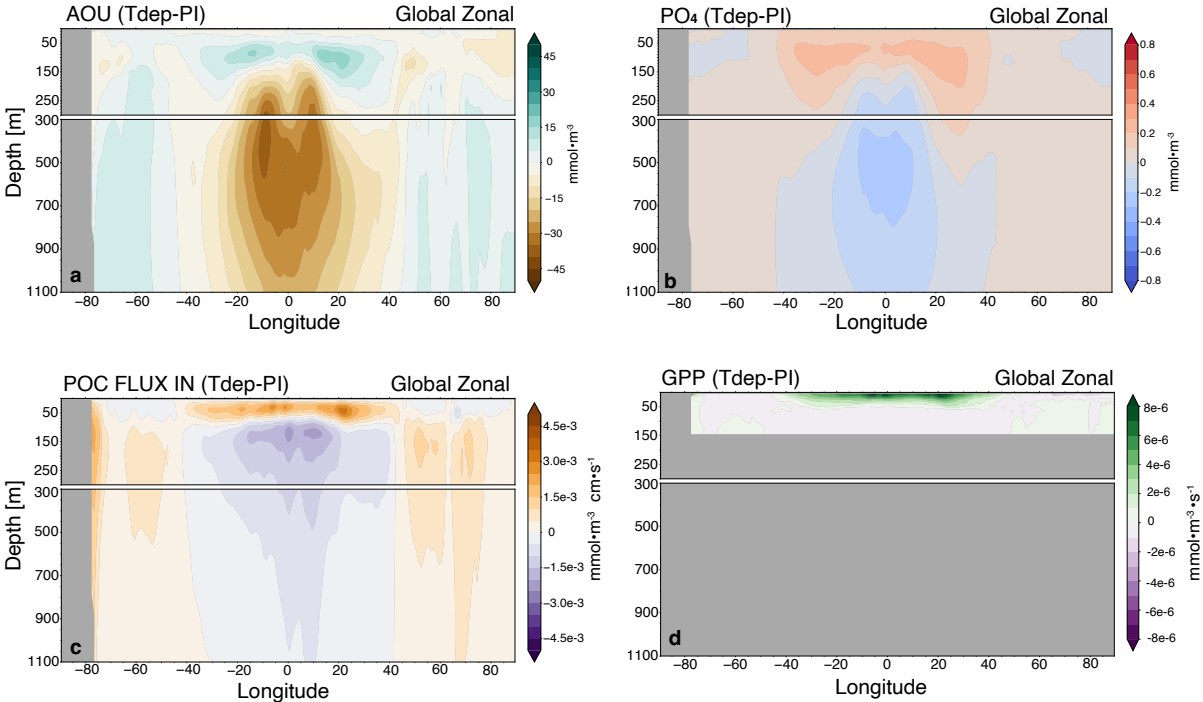

**Figure 12. Global zonal average versus depth.** Shown for a) apparent oxygen utilization (AOU), b) phosphate, c) particulate organic carbon flux (POC_FLUX_In), and d) gross primary productivity (GPP). All are shown as differences between Tdep and preindustrial (PI) control.

Overall, transfer efficiency in the cool high latitudes has a greater sensitivity to the introduction of temperature-dependent remineralization, suggesting an enhanced high latitude control for carbon sequestration, with the caveat of being offset by

enhanced upper ocean nutrient recycling and less efficient export in the tropical latitudes. Additional recycling of upper ocean nutrients across the tropics results in higher rates of primary productivity, yet does not drawdown these nutrients as remineralization keeps pace with the increased productivity. This could result from a more shallow mixed layer in these low latitudes and subsequent 'trapping' of nutrients in the upper ocean, while it could also suggest the need for improved parameterization of sinking particles (e.g., size and aggregation) to account for feedbacks in these warm, high productivity regions. These results highlight the need for further investigation into the regional extent and net global impact on the marine carbon cycle of these disparate responses, particularly in warming scenarios with decreased meridional sea surface temperature gradients.

The attenuation of particulate organic carbon is in broad agreement with measured values, and in several cases are improved versus the control simulation (e.g., the Equatorial and Northern Pacific). As with transfer efficiency, high productivity regions off the coast of Peru and in the southern portion of the Arabian Sea show less agreement and could benefit from further tuning and additional parameterizations, such as particle size and aggregation, which can both impact sinking rates.

Enhanced upper ocean remineralization in the tropical bands results in a modest increase in phosphate content of the upper ocean of up to around 0.25 mmol/m$^3$ for most of the region, with highest levels in the warmest water of the western warm pool in the Equatorial Pacific and central Equatorial Indian, while high latitudes see little to no change in upper ocean nutrient content between the control and Tdep simulations (Fig. 10). Overall, both the control and Tdep simulations correlate well with empirical data from World Ocean Atlas (WOA23), with global $R^2$ values around 0.8, although the cRMSE was slightly better for the control simulation versus Tdep by less than a modest 0.05, with the majority of the global difference is driven by the Pacific Ocean signal (Fig. 9). Although the negative nutrient bias between the PI control and Tdep simulations is similar, it appears the majority of the Pacific basin offset arises from the positive nutrient bias in the low latitudes, particularly in the cold-tongue high productivity region in the central and eastern equatorial Pacific, where the inclusion of Tdep enhances the ability of the upper ocean to recycle nutrients more efficiently. In particular, this enhanced nutrient recycling could play a distinct role in sustaining primary productivity and carbon export at the low latitudes in future warming scenarios by replacing dependence on nutrient delivery from Southern Ocean mid-depth water to one that is locally derived through more efficient remineralization (Rodgers et al., 2024). These trends highlight the need for further evaluation of additional parameters influencing remineralization and export such as Oxygen limitation, community structure, and aggregation, particularly in high productivity regions.

This newly integrated Tdep parameterization improves model sensitivity to local temperature conditions and adds a key feedback on the marine carbon cycle that should improve model predictions of the magnitude and direction of ocean feedbacks in warm climate scenarios. The natural progression of this work is to test past warm climates to better gauge model performance relative to paleoproxy records, which is currently in progress. Code modifications made here should also be portable to the newer ecosystem module MARBL being utilized in CESM2. Future optimization of the marine carbon cycle should include oxygen limitation within the existing code to improve model performance in high productivity regions, as well as a new parameterization scheme for particle size and aggregation of sinking particulate organic matter.

**5 Conclusions**

Here, a new temperature-dependent remineralization scheme has been added to the ecosystem module of CESM 1.2.2 resulting in improved regional patterns of transfer efficiency reflecting response to local temperature conditions while also preserving the overall net global transfer efficiency. Inclusion of Tdep shifts the regional control of carbon export to higher latitudes and

upwelling regions, making them more significant in predicting future carbon cycle response to warming. Results reported here stress model sensitivity to the magnitude and direction of regional response within the biological pump to warming and the potential for shifts in regional controls on the ability of the ocean to take up and store atmospheric carbon dioxide. Further evaluation of these effects using these code modifications are being conducted in the context of the warm Pliocene and will be reported soon as a subsequent publication.

**Code & Data Availability**

Newly formulated model code and data compilation are available on Zenodo at https://doi.org/10.5281/zenodo.16748201 (Brabson et al., 2025). World Ocean Atlas 2023 data are publicly available at https://www.ncei.noaa.gov/products/world-ocean-atlas (Reagan et al., 2023). Post-processing resources utilized for analysis and figure generation are available on Zenodo at https://doi.org/10.5281/zenodo.16794698 (Brabson et al., 2026).

**Author Contribution**

EKB developed new model code with feedback from all authors. NJB, RPA, and LFD completed model setup and ran all simulations with the new Tdep code modifications in both low-resolution (NJB and LFD) and high-resolution (RPA) CESM configurations. Figures and manuscript were generated by EKB and reviewed and edited by all co-authors. PMH and NJB provided financial support for the project.

**Competing Interests**

The authors declare that they have no conflict of interest.

**Acknowledgements**

We would like to acknowledge high-performance computing support from the Derecho system (**doi:10.5065/qx9a-pg09**) provided by the NSF National Center for Atmospheric Research (NCAR), sponsored by the National Science Foundation.

**Financial Support**

This research was partially supported by the National Science Foundation (Grant Numbers: OCE-2404413 & OCE-2402414).

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
