# Peer review of "A Revised Temperature-Dependent Remineralization Scheme for the Community Earth System Model (v1.2.2)"

_EGUsphere, 2025_

## Author Comment (AC1)

**GENERAL COMMENTS**

The manuscript titled "A Revised Temperature-Dependent Remineralization Scheme for the Community Earth System Model (v1.2.2)" presents a comprehensive evaluation of a new temperature-dependent remineralization scheme and its impacts on the global spatial patterns of POC transfer efficiency and nutrient distributions. The authors demonstrate that the new scheme improves the latitudinal contrasts of transfer efficiency and phosphate levels across ocean basins. By validating against in-situ observational profiles and comparing the previous and revised versions of CESM, the study successfully enhances model performance and provides valuable implications for future projections of ocean biogeochemistry. Furthermore, the authors suggest that this new scheme can be applied to future warming scenarios to yield more accurate projections that were previously uncertain.

While the manuscript provides a detailed description of the methodology and results of the temperature-dependent schemes, several key issues should be addressed before publication. In particular, the study should engage more thoroughly with previous research on remineralization schemes, clearly articulate its central research question, and emphasize the novel contributions and distinctions from prior studies to strengthen its scientific impact.

Dear RC1,

Thank you for the positive feedback! All comments are addressed below sequentially.

Best Regards,

Liz, on behalf of all co-authors

**MAJOR COMMENT**

First, the authors should discuss the previously developed remineralization schemes proposed by other ocean biogeochemical modeling groups. For instance, the IPSL, GFDL, and CSIRO groups implemented temperature-dependent remineralization schemes in their models more than a decade ago (Oke et al., 2013; Aumont et al., 2015; Laufkötter et al., 2017; Stock et al., 2020), as also noted in the Methods section of this manuscript. Therefore, the temperature-dependent remineralization scheme presented here cannot be considered entirely new. However, the manuscript lacks a proper introduction and discussion of these earlier schemes. The authors should describe these prior approaches, clearly state how their implementation differs from previous models, and highlight the specific novelty and advancement of this work relative to earlier studies.

- \*\* You are quite correct that there are a number of existing temperature-dependent remineralization schemes introduced across various modeling groups. Our goal with this work was not to introduce temperature-dependent remineralization as a new concept into BGC modules, but rather to implement it as a new parameterization into a widely used earth system model (CESM) where it was previously not present.
- \*\* In terms of prior approaches to temperature-dependent remineralization, a full review is beyond the scope of this current paper. However, others have conducted such reviews, and we agree that additional discussion of the general strategies and success of these other approaches would help to frame our current work. This additional discussion will be added to the introductory material accordingly.

Implementation of this scheme within CESM alone would not be sufficient to justify publication in Geoscientific Model Development (GMD) without a more explicit demonstration of its scientific innovation and contribution.

\*\* With the primary goal of Development and Technical Papers at GMD being to "... describe technical developments relating to model improvements such as the speed or accuracy of numerical integration schemes as well as new parameterizations for processes represented in modules.", we would contend that development and validation of this new parameterization within CESM does demonstrate the innovation and contribution necessary for publication in this journal.

Additionally, despite the detailed description of the new remineralization formula, it remains difficult to directly compare it with the previous remineralization formulation in BEC. It appears that the primary difference lies in the inclusion of  $k_{temp,POC}$  within the decayPOC term. I recommend that the authors clearly describe the previous remineralization scheme and specify what has been improved in the new formulation. For instance, providing a table or schematic that illustrates the previous scheme in black lines and the revised scheme in red lines would greatly help readers visualize and compare the changes. Including other temperature-dependent schemes from different models in the same comparison table would make this section even more informative.

- \*\* A description of the old and revised schemes is provided in Section 2.2, including Equations 1, 2, and 3 for further clarification. You are correct that the main difference is the introduction of the  $K_{tempPOC}$  function (Equation 4), a new base scalelength of 150 m, and new banding of the scalelength parameter. For clarification and transparency, these scalelength values will be added as a new Table with a reference to the corresponding previous values.
- \*\* In terms of a formula-to-formula comparison with other model approaches, this would also be beyond the scope of this current paper given the broad range of formulations and approaches as noted above. A discussion of generalized approaches will be added as noted above.

The study by Rodgers et al. (2024, Nature) should be cited to highlight the importance of temperature-dependent remineralization schemes for projecting future changes in primary production under global warming. In that study, the largest contrast in projected production among models arose between the temperature-dependent IPSL model and the temperature-independent CESM model. Therefore, incorporating the authors' new temperature-dependent scheme could potentially alter the direction of projected production trends. Although the CMIP6 simulations are based on CESM2 rather than CESM1, it is noteworthy that CESM1-BEC also lacked a temperature-dependent remineralization scheme. Hence, it would be valuable to examine how the inclusion of this scheme affects future projections, as mentioned in the manuscript.

\*\* Thank you for this helpful insight. A discussion of the results and implications of the Rodgers et al., (2024) other associated references will be incorporated.

Beyond discussion, I strongly recommend that the authors include at least one future projection experimentsuch as an SSP5-8.5 scenario or a simple 1pctCO2 scenario simulations-to compare with previous studies and to provide a more concrete implication of the temperature-dependent scheme.

\*\* This is a fair addition for consideration, and we agree that these are important next-steps generally used for testing new model code such as this. We are taking a slightly different approach, as we are currently developing and running CESM simulations that utilize warm Pliocene boundary conditions that will be

published within the coming year as a follow-up to this publication. We are using these Pliocene simulations as a test case to examine the response of the marine carbon cycle to the newly implemented temperature-dependent remineralization and are conducting data-model comparisons as a method of validation (to include upper ocean column reconstructions of  $\delta^{13}C$  and  $\delta^{11}B$  [pH], among others). We will add comments to the current manuscript that clarify these next steps and how they are linked with the current paper.

The figures throughout the manuscript require further modification and improvement. For example, consistency should be ensured between filled and open circles representing the PI and Tdep experiments in Figures 9 and 11. In Figure 9, PI is shown with open markers and Tdep with filled ones, whereas in Figure 11, the labeling appears reversed—PI as open and Tdep as filled markers. Consequently, it is unclear whether the authors intended to show that the new scheme improves RMSE and R² for upper-ocean phosphate. However, Figure 9 appears to suggest the opposite trend, with R² decreasing and RMSE increasing from PI to Tdep experiments. If this discrepancy results from simulation errors or mislabeling, it should be clarified in the text or figure captions.

\*\* Your observations are very much appreciated. This labeling error was on behalf of the primary author and will be fixed accordingly ensuring consistency across all figures for color, symbol, and notation.

Additionally, more detailed information should be provided in the Methods section. The manuscript states that the pre-industrial control simulation was used for validation against observations and for comparison between the two remineralization schemes, but details are missing. Specifically, the authors should indicate the total simulation length, the integration period used in the figures. If fully coupled simulations were used, equilibrium may not have been reached, leading to year-to-year variability in climatological fields. In such cases, uncertainty ranges—such as those shown in Figure 6—should be included. The authors report variations in transfer efficiency and latitude-dependent changes, but it remains unclear whether these differences reflect genuine mechanistic responses to the new scheme or simply internal climate variability. Therefore, an estimation of uncertainty, for example using interannual standard deviations, is strongly recommended to clarify these features.

\*\* Thank you also for these observations. The simulation length was 300 years for both the control and Tdep simulations, as noted in Section 2.1, and these were branched from year 2500 of a previously equilibrated run. Also noted in Section 2.1, we believe this is sufficient for quasi-equilibrium, although should full overturning scales be investigated, this would be reconsidered. Results for all figures were averaged over the final 30 years of the simulation in order to eliminate the challenges that you have noted with internal variability. Clarification of the 30 year integrated average will be explicitly added to Section 2.3.

---

## Author Comment (AC2)

**Anonymous Referee #2, 22 Oct 2025**

The manuscript by Brabson et al. describes the implementation of a temperature-dependence of the remineralisation of particulate organic carbon in the CESM1 model. The authors conclude that the new temperature dependence results in an improvement of the transfer efficiency in different temperature regimes of the ocean. The study is interesting and overall clearly presented, with some details missing (see comments below). I recommend it for publication after the following points are addressed:

Dear RC2,

Thank you for the positive feedback! All comments are addressed below sequentially.

Best Regards,

**Liz, on behalf of all co-authors**

I recommend to add a description of the actual values of newly introduced parameters, e.g. the tuning result, for the Tdep parameters (e.g. in a table), and ideally also show a plot of the temperature influence on the remineralisation rate.

\*\* We can certainly add a table of values, which would be primarily for the adjusted scalelength depths (both base scalelength and depth bands). The other value for the temperature parameter was newly added, so it is only the referenced value from the text. For our formulation, we do not calculate remineralization rate directly, but it is instead determined using mass balance as in Equation 2.

Methods: More information on the simulations performed, including information on the tuning process, would be informative in this section.

- \*\* Additional details on the specific details will be added to Section 2.2.
- 1 179ff: it would be good to list the number of data points used for comparison somewhere, along with an exact computation of the error metrics used.
- \*\* Individual data points are shown as triangles in Figure 7. Cite-specific citations for error handling are listed in Table 1. Here, data from each location were normalized to flux at 150 meters for equivalent comparison across the compilation.
- 1. 5: The figure size should be enlarged to enhance readability of the figure inlet.
- \*\* Good point. We will adjust accordingly for easier readability.
- 1. 250: "negative biases of 0 to 1.0" please provide the unit.
- \*\* We will add the units here.
- 1. 255: Why does the Indian and Arctic Ocean perform almost identical with respect to the phosphate concentration, when they have the highest differences in the basin-scale efficiency (Fig. 6)? A sentence addressing this would be helpful.
- \*\* This is a great point, and we will include additional discussion of these results.

Also 1 255.: I don't agree with the statement that Fig. 9 shows a decrease in cRMSE and an increase in R2 for most of the simulations. If the filled symbols indicate indeed the PI simulations, they are all higher for R2 and lower for cRMSE, except for the Indian Ocean and the Arctic. Please doublecheck the symbol description in Fig. 9 (also in comparison to Fig. 11, see comment below), and if it is correct, adjust the statement in 1.255 accordingly. (Same for supplementary figures). For now, I assume this is a plotting mistake, otherwise, some of the conclusions need to be revised.

\*\* You are correct, this is a labeling error. Many apologies. This will be corrected, and all figures double-checked.

Fig. 8: The model performance with respect to the WOA2023 phosphate concentration looks slightly worse in the Tdep simulations in the low latitudes when comparing the PI simulations. This should be addressed in the discussion – why is that the case? Could other processes or controlling factors be missing? Have other parameter values overcompensated the temperature effect before, and require adjusting now that the temperature dependence is included?

\*\* There are definitely other factors at play, specifically in high-productivity regions. We will add further discussion of this, as well as proposed recommendations for next steps to address these challenges.

Figs. 11 and 9: For consistency, I would advise to use filled symbols for either the PI or the Tdep simulation, and not change between the meaning in different figures. At the moment, Fig. 11 has filled symbols for Tdep, and Fig. 9 has filled symbols for PI.

\*\* This will be revised and made consistent.

I could not access the data sets provided in the accompanying link for the review process, perhaps due to an embargo before publication. I can therefore not make any statement on the supplementary datasets.

\*\*We have been able to access on our end. Please let us know if this persists.

---

## Referee Report (RR1)

**GENERAL COMMENTS**

The manuscript titled "A Revised Temperature-Dependent Remineralization Scheme for the Community Earth System Model (v1.2.2)" has been revised in response to the reviewers' comments. However, the revisions are limited, and the manuscript still requires substantial improvement and scientific robustness before it can be considered for publication in my opinion. Several issues previously raised remain unaddressed. I provide additional comments below.

**MAJOR COMMENTS**

1. In Figure 9, as noted previously, the labels for PI and Tdep remain unchanged. The scatter corresponding to PI still shows higher $R^2$ and lower cRMSE, which contradicts the conclusions stated in the manuscript. This discrepancy needs to be clarified; otherwise, the results suggest that the skill of the new remineralization parameterization is worse than that of the previous scheme.

In addition, Figure 9 is not well suited for evaluating model performance, as the two metrics shown are independent of each other. Instead, I recommend using a Taylor diagram, which is a standard tool for assessing climate model performance in terms of correlation, root-mean-square error, and the ratio of variances (Taylor, 2001).

Taylor,K.E.(2001). Summarizing multiple aspects of model performance in a single diagram. JGR: Atmosphere, 106 (D7), 7183–7192

2. In Figure 11, the authors average over the IAO region, which appears to combine the Indian Ocean and the Arctic Ocean. Is there a specific reason for grouping these two regions into a single category? These regions represent very different oceanic environments—for example, the Indian Ocean is a warm, predominantly tropical basin, whereas the Arctic Ocean is a cold, polar system.

In addition, the authors state that the regional definitions follow Weber et al. (2016); however, that study does not combine the Indian and Arctic Oceans into a single region. Finally, even if the ocean-region names are adopted from Weber et al. (2016), it is strongly recommended that all abbreviations be explicitly defined (as was done for AAZ and ETP) to avoid ambiguity.

3. The authors stated in their response that clarification regarding the use of the last 30 years would be provided in Section 2.3; however, the manuscript does not currently include such an explanation. As I understand it, the analysis involves interannual variability, with results presented as 30-year averages. Nevertheless, climate models inherently exhibit year-to-year variability arising from internal variability and model-specific characteristics, particularly in CESM. I therefore recommend that the associated uncertainties be explicitly represented by showing the ranges of interannual variability, for example using standard deviations. Specifically, Figures 5 and 6 could include latitude-dependent shading to indicate variability ranges, and Figure 11 could present uncertainty ranges (e.g., error bars) for each bar.

4. Finally, Figure 7 requires additional clarification by providing quantitative performance metrics, such as the RMSE between PI and Tdep for each location. Aside from the two Equatorial Pacific regions, PI appears to show better agreement with observations at several sites (e.g., ALOHA, Peru, Arabian Sea). Therefore, further information is needed to clearly demonstrate that the new temperature-dependent remineralization parameterization represents a genuine improvement over the previous formulation, rather than a degradation in model performance.

---

## Author Response (AR2)

Author Response to Anonymous Referee #1
Submitted on 14 Jan 2026

Dear Anonymous Referee #1,

Thank you for your thoughtful feedback. We have addressed all comments below sequentially.

Best Regards,
Liz, on behalf of all co-authors

**The manuscript titled "A Revised Temperature-Dependent Remineralization Scheme for the Community Earth System Model (v1.2.2)" has been revised in response to the reviewers' comments. However, the revisions are limited, and the manuscript still requires substantial improvement and scientific robustness before it can be considered for publication in my opinion. Several issues previously raised remain unaddressed. I provide additional comments below.**

**MAJOR COMMENTS**

**1. In Figure 9, as noted previously, the labels for PI and Tdep remain unchanged. The scatter corresponding to PI still shows higher R2 and lower cRMSE, which contradicts the conclusions stated in the manuscript. This discrepancy needs to be clarified; otherwise, the results suggest that the skill of the new remineralization parameterization is worse than that of the previous scheme.**

Thank you for noting the Figure 9 labels. In our previous response, we suggested that the labels could have been reversed. However, upon review, we confirmed that they were correct and that no change was necessary. This clarification is now stated explicitly in the revised manuscript in the Figure 9 caption comments.

The objective of the temperature-dependent remineralization (Tdep) implementation was to maintain upper-ocean nutrient concentrations comparable to the control simulation while improving transfer efficiency as the primary metric. Figure 9 shows that the PI control exhibits slightly lower cRMSE and higher $R^2$ than the Tdep simulation. However, the magnitude of the cRMSE difference is comparable to the characteristic inter-product mismatch between WOA and GLODAP for global phosphate concentrations, reported to be ~0.03 for the full water column. Differences in cRMSE of this magnitude are therefore interpreted as marginal and within bounds of the observational product disagreement. Clarification of this point was added in line 285.

Similarly, $R^2$ values for the Atlantic, Southern Ocean, and Indian Ocean are nearly identical between configurations, while the global mean $R^2$ differs by ~0.02, which is considered a small difference in pattern agreement. The largest $R^2$ difference occurs in the Pacific basin (~0.1). This deviation is considered meaningful and is addressed explicitly in expanded Discussion (~line 376) on the role of tropical nutrient distributions and regional sensitivity.

**In addition, Figure 9 is not well suited for evaluating model performance, as the two metrics shown are independent of each other. Instead, I recommend using a Taylor diagram, which is a standard tool for assessing climate model performance in terms of correlation, root-mean-square error, and the ratio of variances (Taylor, 2001).**

**Taylor,K.E.(2001). Summarizing multiple aspects of model performance in a single diagram. JGR: Atmosphere, 106 (D7), 7183–7192**

We agree that Taylor diagrams are a standard and widely used tool for evaluating climate model performance (Taylor, 2001). The metrics shown in Figure 9 (cRMSE and $R^2$) correspond to two of the three components summarized in a Taylor diagram. While these metrics diagnose different aspects of model performance, they are not statistically independent, as cRMSE is mathematically related to correlation and standard deviation within the Taylor framework. We therefore interpret cRMSE and $R^2$ as complementary diagnostics of model accuracy and pattern agreement.

We intentionally did not include the standard deviation ratio, as it is not central to the scientific question addressed here. Our focus is on model–data agreement in upper-ocean nutrient distributions and residual error magnitude, for which cRMSE and $R^2$ provide a sufficient and appropriate assessment of model performance.

**2. In Figure 11, the authors average over the IAO region, which appears to combine the Indian Ocean and the Arctic Ocean. Is there a specific reason for grouping these two regions into a single category? These regions represent very different oceanic environments—for example, the Indian Ocean is a warm, predominantly tropical basin, whereas the Arctic Ocean is a cold, polar system.**

**In addition, the authors state that the regional definitions follow Weber et al. (2016); however, that study does not combine the Indian and Arctic Oceans into a single region. Finally, even if the ocean-region names are adopted from Weber et al. (2016), it is strongly recommended that all abbreviations be explicitly defined (as was done for AAZ and ETP) to avoid ambiguity.**

The regional definitions used in Figure 11 were taken directly from the basin mask files provided by Thomas Weber. You are correct that the IAO region combines the Indian Ocean and the Arctic Ocean. In Weber et al. (2016), this combined region is defined in the basin mask files but is excluded from their subsequent analysis (also noted in personal correspondence with Thomas Weber), with the focus placed instead on the remaining eight regions.

While we acknowledge that the Indian and Arctic Oceans represent very different oceanographic environments, we chose to retain the IAO region in this analysis in order to provide a complete, global assessment of basin-scale responses. Including this region allows us to evaluate whether trends observed elsewhere, particularly improvements in transfer efficiency with the inclusion of temperature-dependent remineralization, are also evident when considering the full global ocean. We have clarified this choice in the revised text (line 326) to avoid ambiguity regarding consistency with Weber et al. (2016).

We agree that explicit definition of all acronyms is important for clarity. A complete list of regional abbreviations has now been added to the Figure 11 caption, consistent with the definitions provided elsewhere in the manuscript (e.g., AAZ and ETP).

**3. The authors stated in their response that clarification regarding the use of the last 30 years would be provided in Section 2.3; however, the manuscript does not currently include such an explanation. As I understand it, the analysis involves interannual variability, with results presented as 30-year averages. Nevertheless, climate models inherently exhibit year-to-year variability arising from internal variability and model-specific characteristics, particularly in CESM. I therefore recommend that the associated uncertainties be explicitly represented by showing the ranges of interannual variability, for example using standard deviations. Specifically, Figures 5 and 6 could include**

**latitude-dependent shading to indicate variability ranges, and Figure 11 could present uncertainty ranges (e.g., error bars) for each bar.**

Apologies for the confusion regarding the previously stated addition of 30 year averages in Section 2.3. This clarification was actually added in Section 2.1 (Line 94) in the previous track-changes version.

The analyses presented here are intended to characterize equilibrium or climatological differences between model configurations rather than interannual variability. For this reason, we use the 30-year average to suppress seasonal and year-to-year fluctuations and to represent the equilibrated mean state. This averaging period is standard practice for CESM equilibrium analyses and is sufficient to isolate the signal of interest. However, visualizing the range of interannual variability captured in these averages is a fair concern. To address this, we have added a 1-$\sigma$ band to the global zonal TE in Figure 5, and commented further in both the figure caption text and main text noting the addition.

Figure 11 depicts the residuals of TE in PI and Tdep vs those reported in Weber. Since the Weber product is statistically derived based on both empirical data and modeling products, these data do have interannual variability for comparison.

**4. Finally, Figure 7 requires additional clarification by providing quantitative performance metrics, such as the RMSE between PI and Tdep for each location. Aside from the two Equatorial Pacific regions, PI appears to show better agreement with observations at several sites (e.g., ALOHA, Peru, Arabian Sea). Therefore, further information is needed to clearly demonstrate that the new temperature-dependent remineralization parameterization represents a genuine improvement over the previous formulation, rather than a degradation in model performance.**

We appreciate the reviewer's comment. As mentioned in response to #1 above, the primary goal of this analysis was to improve transfer efficiency performance with the inclusion of temperature-dependent remineralization (Tdep), and this is the main metric that was targeted for improvement. Following this, POC flux attenuation and upper-ocean nutrient content (noted above) were used as secondary diagnostics to confirm that the model behavior remained largely consistent in both the control and Tdep simulations. While transfer efficiency improves at some sites, such as in the Equatorial Pacific, other regions like Peru show more modest differences, as noted ~ line 266.

We used methodology following Laufkötter et al. (2017) for the POC flux attenuation analysis, applying it to an expanded data compilation. This approach is a widely accepted benchmark for assessing model performance, and although there are alternative methods, we chose this one due to its established use in similar studies for validating the behavior of biological carbon pump models.

Author Response to Anonymous Referee #2
Submitted on 14 Jan 2026

Dear Anonymous Referee #2,

Thank you for your thoughtful feedback. We have addressed all comments below sequentially.

Best Regards,
Liz, on behalf of all co-authors

1. **In the response letter, the authors mentioned additional discussion on why the flux attenuation in the Southern Ocean (Fig. 7) is almost identical in both PI and Tdep, while their transfer efficiency in Fig. 6 is vastly different. This difference is noted in the manuscript (line 275), but the discussion would benefit from a few more details. I suggest adding an explanation for this discrepancy.**

Thank you for highlighting this important point. Indeed, there is a marked difference in the transfer efficiency of the Southern Ocean region, as shown in Figure 6. A key detail is with the depth bands used for the transfer efficiency metric, which is at a fixed 100 meters and 1000 meters. Looking at the flux at only 1000 meters for the Southern Ocean site, the modeled PI flux is lower (around 25%) than the Tdep (around 30%). Given similar export at 100 meters, this would account for the higher transfer efficiencies. This subtlety, related to the 100 meter and 1000 meter fixed depths used by the community for TE, is something that we feel is an area of further discussion and potential for development of a new standardized metric for TE that can account for upper column (0-100m) adjustments of the pump (see e.g., line 355), perhaps by considering integrated depth bands versus an individual depth similar to discussed in Buesseler et al., 2020.

In terms of consistent trends in cold water regions for both TE and POC flux as you note, we have also added further discussion in the body of the text (line 266).

2. **Similarly, the discussion regarding Fig. 8 (see previous comment) would also benefit from more details: I recommend to include 1-2 sentences in the discussion about \*why\* the phosphate concentration in the Tdep simulation does not perform as well as in the PI simulation despite the improvements in simulating transfer efficiency. This is an important point for potential future improvements.**

This is a great point and a topic of further discussion. We have added additional discussion (line 376).